# Successful Aging among Immigrant and Canadian-Born Older Adults: Findings from the Canadian Longitudinal Study on Aging (CLSA)

**DOI:** 10.3390/ijerph192013199

**Published:** 2022-10-13

**Authors:** Mabel Ho, Eleanor Pullenayegum, David Burnes, Esme Fuller-Thomson

**Affiliations:** 1Factor-Inwentash Faculty of Social Work, University of Toronto, Toronto, ON M5S 1V4, Canada; 2Institute for Life Course & Aging, University of Toronto, Toronto, ON M5S 1V4, Canada; 3Child Health Evaluative Sciences, The Hospital for Sick Children Research Institute, Toronto, ON M5G 0A4, Canada; 4Dalla Lana School of Public Health, University of Toronto, Toronto, ON M5T 3M7, Canada

**Keywords:** Canadian Longitudinal Study on Aging (CLSA), older adults, immigrants, immigration, successful aging

## Abstract

Background: Few studies in Canada have focused on the relationship between immigrant status and successful aging. The concept of successful aging used in this study includes the ability to accomplish both activities of daily living (ADLs) and instrumental activities of daily living (IADLs), freedom from mental illness, memory problems and disabling chronic pain, adequate social support and older adults’ self-reported happiness and subjective perception of their physical health, mental health and aging process as good. Methods: The present study analyzed the first two waves of data from the comprehensive cohort of the Canadian Longitudinal Study on Aging (CLSA). The sample includes 7651 respondents aged 60+ at time 2, of whom 1446 respondents were immigrants. Bivariate and multivariable binary logistic regression analyses were conducted. Results: Canadian-born older adults had a slightly higher prevalence and age-sex adjusted odds of achieving successful aging than their immigrant counterparts (aOR = 1.18, 95% CI: 1.04, 1.34, *p* < 0.001). After adjusting for 18 additional factors, immigrant status remained statistically significant (aOR = 1.24, 95% CI: 1.09, 1.41, *p* < 0.001). Significant baseline factors associated with successful aging among immigrants included being younger, having higher income, being married, not being obese, never smoking, engaging in moderate or strenuous physical activities, not having sleeping problems and being free of heart disease or arthritis. Conclusions: Immigrant older adults had a lower prevalence of successful aging than their Canadian-born peers. Further research could investigate whether policies and interventions supporting older immigrants and promoting a healthy lifestyle enhance older adults achieve successful aging in later life.

## 1. Introduction

The term “successful aging” was first introduced in the 1960s by Havighurst [1], who stated, “in order to provide good advice, it is essential that gerontology have a theory of successful aging” [1] (p. 8). For the following six decades, thousands of conceptual and empirical articles have tried to define successful aging and how it can be achieved [2,3]. Most research studies on successful aging have been conducted in the United States, Asia and Europe, while relatively few studies have been published in Canada (for important exceptions, please see [4,5]). The concept of successful aging used in this study includes adequate social support, the ability to accomplish both ADL and IADL activities, freedom from mental illness, memory problems and disabling chronic pain, in addition to the older adults’ self-reported happiness and subjective perception of their physical health, mental health and aging process as good.

Although age-related chronic and degenerative conditions may not be avoidable, people can develop and implement age-friendly policies and interventions to support older adults to age well. If health and social systems are better aligned with the needs of the older adults [6,7] and focus on their strengths rather than their deficits [8,9], more older adults can enjoy a vibrant and successful later life. Using the first two waves of data from a large, national, longitudinal study on aging, the Canadian Longitudinal Study on Aging (CLSA) [10], the present study examined the role of immigrant status in the context of successful aging.

With improved knowledge of successful aging, policymakers, researchers and social workers, in collaboration with older adults and their families, can develop better policies and interventions, conduct relevant research and advance knowledge concerning the trajectories of successful aging. With such improved knowledge, Canada will be more prepared in how best to provide conditions in which older adults can genuinely thrive as they age and experience a high quality of life.

### 1.1. Background

**Successful Aging.** There has been an effort to shift gerontology from a disease-focus toward a wellness-orientation, using the concept of “successful aging.” The present study has chosen successful aging as the guiding notion for three reasons. Firstly, it has a relatively long history compared to alternative concepts of aging well, such as “healthy,” “productive,” and “positive” aging [11] (p. 3). Secondly, it is the dominant and most frequently referenced conceptual framework in the study of aging [12]. Finally, it provides a relatively broader understanding of aging by covering the physiological, psychological and social aspects [13]. Definitions of successful aging come from two main sources: researcher-defined classifications of successful aging and lay perspectives of older adults on successful aging. However, there is still no consensus definition of successful aging and how it can be achieved [14,15,16]. The different classifications of successful aging defined by researchers define approximately one-third of older adults as successful agers [17]. In sharp contrast, most middle-aged and older adults (92–94%) perceive themselves as successful agers when asked to rate themselves [18,19].

In the study of Montross and colleagues [19], although 92% of older adults in their study rated themselves as aging successfully, only 5% met researcher-defined classifications of successful aging. This finding is consistent with studies on older adults’ perceived successful aging and late-life disability [20]. This discrepancy is caused because researcher-defined classifications of successful aging often focus on physical wellness and exclude those who have any chronic disease and physical disability. According to the Canadian Health Survey on Seniors [21], many older adults live with chronic diseases: 45% with hypertension, 38% with arthritis, 22% with diabetes, 19% with heart disease and 5% with osteoporosis. The Canadian Survey on Disability [22] also reports that 47% of those aged 75 and over have one or more disabilities. Studies also show that households with people with disabilities require on average a higher level of income to meet the same standard of living compared to households without people with disabilities [23] and there is a positive relationship between disability and poverty [24]. Thus, researcher-defined classifications of successful aging requiring the absence of physical illness and disability do not reflect the reality of older adults’ lived experiences.

Among various models of successful aging, Young et al.’s [25] multidimensional model of successful aging examines the individual trajectories of successful aging and pays attention to structural issues affecting the individuals. According to Young et al. [25] successful aging is defined as “a state wherein an individual is able to invoke adaptive psychological and social mechanisms to compensate for physiological limitations to achieve a sense of well-being, high self-assessed quality of life and a sense of personal fulfilment even in the context of illness and disability” [25] (pp. 88–89). This expanded view on successful aging addresses the discrepancies between researcher-defined classifications of successful aging and lay perspectives of older adults on successful aging.

**Immigrant Status.** Canada has long been a land of immigrants and it continues to welcome hundreds of thousands of immigrants every year [26]. Canada’s immigrant population comprises 21.9% (7,540,830 people) of the Canadian population and this number is growing [26]. The Government of Canada anticipates welcoming more than 400,000 immigrants each year between 2021 and 2023 [27].

Even though Canada has a large immigrant population, few studies compare successful aging between domestic- and foreign-born older adults. Studies have found a “healthy immigrant effect”—immigrants are generally healthier than domestic-born Canadians when they arrive in Canada [28,29,30]. Immigrants have a considerable advantage over those born in Canada when age-standardized mortality rates are examined for all avoidable causes of mortality [31]. However, studies conducted in the United State have shown that this health advantage erodes over time and is replaced by the “unhealthy assimilation of immigrants”—the longer the immigrants stayed in the new countries, the poorer their health status becomes [32]. For example, the body mass index (BMI) completely converged to native-born BMIs within 10 years of arrivals for women and 15 years of arrival for men [32].

Researchers hypothesize that only people who are healthy decide to migrate and that plays a critical role in explaining the healthy migrant phenomenon [33,34]. Studies have found that older immigrants tend to do better if they have strong social networks (e.g., family, neighbors) [35,36], social participation (e.g., church or religious activities, community programs) [37,38] and access to the internet (e.g., social engagement, leisure activities) [39]. However, studies have also found that recent immigrants may experience psychological distress due to adjustment problems, economic hardships, negative employment experiences, lack of social support and ethnic discrimination [40]. They are also more reluctant to seek professional help [41]. To the best of our knowledge, there is no study on the relationship between immigrant status and successful aging in Canada [42] and a paucity of research studying successful aging among immigrants in the United States (for important exceptions, please see [20,43,44]) constitutes the only literature on this topic.

### 1.2. Conceptual Framework

Aging is a complex blending of biological, behavioral, environmental and social changes [45,46]. However, some studies have often overlooked the influence of the dynamic interplay of individual and environmental factors [47]. As noted above, studies on successful aging focus primarily on the physiological aspects of aging [17], often neglect various structural issues influencing older adults [48] and often employ a classification of successful aging that emphasizes physical/functional health without accounting for the subjective experiences of older adults [49]. Guided by a conceptual framework that synthesizes three theoretical perspectives: ecological systems theory [50,51], a multidimensional model of successful aging [25] and the concept of complete mental health [52], this study defines successful aging as a combination of physical, psychological, mental, social and self-rated wellness, regardless of the presence of physical illness and disability. In other words, successful aging is a state wherein an older adult achieves a sense of physical, psychological, mental, social and self-rated well-being even in the context of chronic health conditions and physical disability.

The concept of successful aging used in this study is built on both objective and subjective measures of aging well. In keeping with many researcher-derived definitions, the new concept includes the absence of memory problems, freedom from any serious mental illness or chronic disabling pain and adequate social support. With respect to physical health, we have modified the usual restriction that successful agers must be free of any chronic health conditions. Instead, our definition requires that individuals have no limitations in ADLs and IADLs, regardless of the number of chronic illnesses that they have. In addition, we have incorporated the older adults’ subjective perception of their aging process, physical health and mental health, as well as their self-reported emotional well-being (e.g., happiness and life satisfaction).

### 1.3. Study Aims

This study examines the relationship between immigrant status and successful aging among older Canadians using the first two waves of data from the Canadian Longitudinal Study on Aging (CLSA). The present study has provided an expanded definition of successful aging as described above. The study investigates the following research questions:Do immigrants have (a) a greater prevalence and (b) higher age-sex adjusted odds of successful aging than their Canadian-born peers in the 2015–2018 wave of the Canadian Longitudinal Study on Aging (CLSA) comprehensive cohort?What baseline factors, if any, attenuate the association between immigrant status and subsequent successful aging?Among immigrants, what baseline characteristics predict successful aging during the follow-up wave?

## 2. Materials and Methods

### 2.1. Study Population

Participants in this study were drawn from the baseline (gathered 2011–2015) and follow-up 1 data (hereafter referred to as time 2 data; gathered 2015–2018) from the CLSA Comprehensive Cohort [53,54]. The CLSA Comprehensive Cohort at baseline consisted of 30,097 Canadian men and women aged 45 and 85 years. Respondents were randomly selected from 7 provinces at baseline. This longitudinal study follows respondents with repeated waves of data collection every three years for at least 20 years or until death. Respondents from the comprehensive CLSA cohort were interviewed face-to-face in their homes and assessed in person at one of the eleven CLSA data collection sites. Respondents underwent additional interviews, assessments, measurements and tests at the data collection sites. Thirteen research ethics boards across Canada approved the research protocol of the CLSA. Further information about the CLSA can be found at www.clsa-elcv.ca (accessed on 28 August 2022).

Of the 30,097 respondents at baseline, 27,799 respondents participated in time 2. Of these, 18,978 met our inclusion criteria of being aged 60 years or older at time 2. In order to calculate the three-year incidence rate of successful aging, 10,375 respondents were excluded because they were not aging successfully at baseline. An additional 714 respondents had missing data on one or more of the variables which were used to create the successful aging variable at time 2 and were, therefore, also excluded. Thus, the final sample size included 7651 respondents, of whom 1446 were immigrants. This study involving secondary data analysis of CLSA data, which was approved by the Health Sciences Research Ethics Board of the University of Toronto (protocol number: 38284).

### 2.2. Measures

#### 2.2.1. Dependent Variables Assessed at Both Baseline and Time 2

This paper’s definition of successful aging was a multidimensional construct which integrated the elements discussed above, using both research-derived and older-adult-derived definitions. It is comprised of four domains: (1) physical wellness, defined as the absence of disabling physical conditions rather than simply the absence of chronic diseases and physical disability, (2) psychological and emotional wellness, (3) social wellness and (4) self-rated wellness on the aging process, physical health and mental health. Respondents who met all four criteria were classified as successful agers. Otherwise, they were considered typical agers. Consistent with previous studies [4,55], the presence of chronic physical illness did not preclude successful aging as long as it did not interfere with daily functioning or cause disabling chronic pain [25].

**Physical Wellness**. As indicated in Table 1, respondents were classified as being physically well if they reported that their physical conditions did not prevent them from performing some of their (1) Activities of Daily Living (ADLs) and (2) Instrumental Activities of Daily Living (IADLs) and they were (3) not having disabling pain and discomfort. ADLs and IADLs were assessed by yes or no questions. ADLs covered five areas: (1) being able to dress and undress oneself without help; (2) being able to eat without help; (3) being able to walk without help or (4) being able to walk with some help from a person or with the use of a walking aid; and (5) being able to get in and out of bed without help or aides. IADLs covered eight areas: (1) being able to use the telephone without help; (2) being able to get to places out of walking distance without help; (3) being able to go shopping for groceries or clothes without help; (4) being able to prepare meals without help; (5) being able to do housework without help; (6) being able to do housework with some help; (7) being able to take medicine without help; and (8) being able to handle money without help. Not having disabling pain and discomfort referred to being usually free of disabling pain and discomfort. In other words, respondents were usually free of pain and discomfort and their pain and discomfort did not prevent them from engaging in some of their activities.

**Psychological and Emotional Wellness**. Respondents reported no issue in the following seven areas were classified as meeting the criteria of psychological and emotional wellness: (1) not having depression as classified by the CES-D score [56,57]; (2) not having a diagnosis of anxiety; (3) not having PTSD as classified by the Primary Care Posttraumatic Stress Disorder (PC-PTSD) score [58]; (4) not always feeling depressed (less than 2 days in the past week); (5) always feeling happy (more than 3 days in the past week); (6) always feeling satisfied with life (more than 3 days in the past week) [59]; and (7) not having a memory problem.

**Social Wellness**. Social wellness was measured by the availability of social support [60]. Respondents were asked if they had (1) someone to give them advice about a crisis; (2) someone to show them love and affection; and (3) someone to confide in or talk to about themselves or their problems. The response options were dichotomized into two levels: (1) none/a little/some of the time; and (2) most/all of the time. Respondents reporting most/all of the time in all three questions were considered as having social wellness.

**Self-rated Wellness**. Self-rated wellness was measured by self-rated aging, physical health and mental health. Respondents rated their perception of (1) own aging, (2) physical health and (3) mental health. The response options were dichotomized into two levels: (1) poor to fair; and (2) good to excellent. Respondents reporting good to excellent in all three questions were considered as having self-rated wellness.

**Successful Aging**. Respondents who met all four criteria of physical, psychological, emotional, social and self-rated wellness were classified as successful agers. Otherwise, they were considered typical agers. Successful aging status was identified at both baseline and time 2.

**Immigrant status**. Immigrant status was measured as a dichotomous (no/yes) variable according to whether a respondent self-identified as being born in Canada or another country.

#### 2.2.2. Covariates

Guided by the conceptual framework discussed above and based on the literature review on factors related to successful aging, within the constraints of the variables available in the CLSA, baseline demographic, socioeconomic, lifestyle and health-related variables that could attenuate the relationship between immigrant status and successful aging were included in the analyses. The covariates included demographic (age, sex, marital status); socioeconomic (education level, wealth measure (i.e., house ownership status) and poverty line status), lifestyle factors (BMI, smoking status, various exercises—sitting activities, walking, light sports, moderate sports, strenuous sports, muscle and endurance exercises and sleep problems); and health-related indicators (i.e., physical diseases). Six different clusters of potential effect modifiers were added sequentially to the model.

**Demographic factors**. There were two clusters of demographic factors. The first cluster included age during the baseline wave in categories (55 to 59/60 to 64/65 to 69/70 to 74/75 to 79/80+) and sex (male/female). This cluster was added to all models, except the first model where only the effect of immigrant status was examined. The second cluster included marital status (single, never married or never lived with a partner/married or living with a partner/common-law relationship/widowed/divorced or separated).

**Education factors**. The cluster of education factors included education level (less than secondary school graduation/secondary school graduate and/or some post-secondary education/post-secondary degree/diploma).

**Lifestyle factors**. The cluster of lifestyle factors and BMI included smoking status, sitting activities, walking, light sports, moderate sports, strenuous sports, muscle and endurance exercises, sleep problems and BMI. Smoking status included never smoked, a former smoker and a current smoker. Sitting activities (never or seldom/sometimes or often) included computer activities, doing handicrafts, reading and watching TV. Walking (never or seldom/sometimes or often) referred to taking a walk outside the home or yard for any reason. Various levels of sports were measured by asking respondents if they had engaged in such activities over the past seven days. Light sports (never or seldom/sometimes or often) included recreational activities such as badminton, bowling, golf with a cart, fishing, shuffleboard, or other similar activities. Moderate sports (never or seldom/sometimes or often) included ballroom dancing, hunting, golf without a cart, skating, softball and alike. Strenuous sports (never or seldom/sometimes or often) included aerobics, cycling, jogging, skiing, snowshoeing, swimming and other similar activities. Muscle and endurance exercises (never or seldom/sometimes or often) included lifting weights, push-ups and alike. Sleep problems (never, rarely, or some of the time/occasionally or all of the time) were measured by self-rated question. BMI was divided into three categories: underweight or normal weight, overweight and obese.

**Physical diseases**. The cluster of physical diseases included yes/no questions that asked if respondents had the following physical diseases: diabetes, heart disease, hypertension, arthritis and osteoporosis.

**Financial well-being**. The cluster of financial well-being included wealth measure (i.e., house ownership) and poverty line status. Wealth measure was categorized into paying rent, paying the mortgage and paying off the mortgage. As indicated in Table 2, poverty line status was calculated by comparing household income and household size with the poverty line in 2015 when the data collection of time 2 began [61] and it was categorized as under poverty line income, marginal income and above poverty line income.

### 2.3. Statistical Analysis

The present study used SPSS Version 28 to conduct all analyses. An adjusted weighting variable was created by dividing the trimmed inflation weights of each unit used in the analysis by the mean of the weights of all analyzed units. A series of multivariable binary logistic regression analyses were conducted using a priori selection criteria (i.e., immigrant status (IS); age and sex; education; marital status; lifestyle factors; physical diseases; wealth measure (i.e., house ownership status) and poverty line status). A significance level of 0.05 (*p* < 0.05) was considered statistically significant for all the tests and 95% confidence intervals (95% CI) were used in the logistic regression. The Hosmer-Lemeshow test was used to test model fit and variance inflation factor diagnosis were employed to assess for multicollinearity.

## 3. Results

### 3.1. Descriptive Statistics

Descriptions of the sample size (unweighted counts), weighted percentages and chi-square statistics based upon the weighted data of the final sample (*n* = 7651) are presented in Table 3. Of these, 97.5% of them responded to the question, “In terms of your own healthy aging, would you say it is excellent, very good, good, fair, or poor?” and rated their aging process as excellent, very good or good.

With the expanded definition of successful aging presented in this study, the prevalence of successful aging at time 2 was 70.5% (95% CI: 0.70, 0.72). The present study identified more than 7 in 10 (72.3%) of those who rated their aging process as good to excellent as successful agers using the newly defined measure of successful aging described above.

### 3.2. Statistical Analysis


**Research Question 1a: Do immigrants have a greater prevalence of successful aging than their Canadian-born peers in the 2015–2018 wave of the Canadian Longitudinal Study on Aging (CLSA) comprehensive cohort?**


Among the 6205 non-immigrant participants, 71.6% were successful agers. Among the 1446 immigrant participants, 65.6% were successful agers. The percentage was also statistically significantly lower for immigrant participants (*x*^2^(1) = 20.6, *p* < 0.001), even though more than three-quarters of the 1446 immigrants in the sample had migrated to Canada at least 4 decades prior to the study. More than half of the non-immigrant participants came to Canada at the age of 18 to 49 (63.4%) and had lived in Canada for more than 40 years (77.2%) (analyses for the duration of immigration are not shown in the table).


**Research Question 1b: Do immigrants have higher age-sex adjusted odds of successful aging than their Canadian-born peers in the 2015–2018 wave of the Canadian Longitudinal Study on Aging (CLSA) comprehensive cohort?**


The results of the binary logistic regression models (see Table 4 and Figure 1) confirmed that the age-sex adjusted odds of achieving successful aging at time 2 were significantly higher among Canadian-born older adults (Model 2: aOR = 1.18, 95% CI: 1.04, 1.34) compared to immigrant older adults.


**Research Question 2: What baseline factors, if any, attenuate the association between immigrant status and subsequent successful aging?**


As shown in Table 4, the unadjusted odds of achieving successful aging were about 22.1% higher for Canadian-born older adults than for immigrant older adults (aOR = 1.22, 95% CI: 1.08, 1.38) when only immigrant status was considered. In the Full Model, which adjusted for 20 factors, the odds of successful aging for Canadian-born older adults were similar (aOR = 1.24, 95% CI: 1.09, 1.41), suggesting that the factors included in the fully adjusted model failed to attenuate the association between immigrant status and successful aging. Other significant baseline factors associated with successful aging in the full sample included being younger, female sex, having higher income (i.e., above poverty line income), being married, not being obese, never smoking, engaging in moderate or strenuous physical activities, not having sleeping problems and living without heart disease or arthritis.


**Research Question 3: Among immigrants, what baseline characteristics predict successful aging during the follow-up wave?**


Among the 1446 immigrants, the results of the binary logistic regressions indicated that the odds of achieving successful aging were higher for respondents who were 55–74 years when compared to those who were 80 years and over (55–59 years: aOR = 2.21, 95% CI: 1.20, 4.05; 60–64 years: aOR = 2.85, 95% CI: 1.65, 4.91; 65–69 years: aOR = 2.67, 95% CI: 1.55, 4.60; 70–74 years: aOR = 2.22, 95% CI: 1.28, 3.87). There was no significant difference between those aged 75–79 years when compared to the 80 years and older group (aOR = 1.21, 95% CI: 0.69, 2.10, *p* = 0.509). There was no statistically significant difference in sex, education, wealth measure (i.e., house ownership status), poverty line status, marital status, BMI, or smoking status.

People who engaged in strenuous sports had higher odds of achieving successful aging (strenuous sports: aOR = 1.47, 95% CI: 1.09, 1.99) compared to those who did not engage in such activities. There was no significant association with successful aging related to engaging in sitting activities, walking, light sports, moderate sports, muscle and endurance exercises, or having sleeping problems.

People who did not have arthritis had higher odds of achieving successful aging (aOR = 1.65, 95% CI: 1.08, 2.52). However, this association was not observed in people who did not have diabetes, heart disease, hypertension, or osteoporosis.

### 3.3. Assessment of Model Fit

For the model including both non-immigrants and immigrants, the results of the Omnibus Tests of Model Coefficients were highly significant (*x*^2^(33) = 372.0, *p* < 0.001), indicating that the final model is significantly better than the baseline model. For the model restricted to immigrants, the results of the Omnibus Tests of Model Coefficients were also highly significant (*x*^2^(32) = 83.6, *p* < 0.001). All variance inflation factors of the predictor variables in both models ranged from 1.00 to 1.22 (VIF < 10), indicating that multicollinearity was not a concern.

## 4. Discussion

This study introduced an expanded measure of successful aging by combining modified researcher-defined classifications of successful aging and lay perspectives of successful aging. The concept of successful aging used in this study includes adequate social support, the ability to accomplish both ADL and IADL activities, freedom from mental illness, memory problems and disabling chronic pain, in addition to the older adults’ self-reported happiness and subjective perception of their physical health, mental health and aging process as good to excellent. Earlier more restrictive researcher-defined classifications of successful aging classified one-third of older adults as successful aging [17]. They were often criticized for requiring the absence of any physical illness and disability. As many older Canadians live with some forms of chronic diseases and physical disability [21,22], it is not realistic to exclude everyone living with chronic health conditions and physical disability from the definition of successful aging. In the present study, the new measure of successful aging considered researcher-defined classifications of successful aging. It incorporated the absence of disabling physical conditions instead of simply the absence of physical disability and illness.

### 4.1. Implications

Older adults whose physical conditions did not prevent them from engaging in some of their activities could still be considered as aging successfully in the definition used in this study of the concept. The expanded definition of successful aging also considered the lay perspectives of successful aging by including the presence of self-rated wellness on the aging process, physical health and mental health, which was considered an important factor by older adults but was often ignored in researcher-defined classifications of successful aging [18,19].

Earlier researcher-derived definitions of successful aging classified one-third of older adults as successful agers [17], but most middle-aged and older adults (92–94%) considered themselves as aging successfully [18,19].

Using the new measure of successful aging presented in this study, the prevalence of successful aging at time 2 was 70.5% (95% CI: 0.70, 0.72), which rose to 72.3% among those who perceived themselves as aging well. Thus, the new measure of successful aging had brought the gap between researcher-definition classifications of successful aging and lay perspectives of successful ager closer. Considering that the expanded definition of successful aging took into account physical wellness, psychological and emotional wellness, social wellness and the self-rated wellness of older adults, we suggest that it is more realistic and better reflective of older adults’ experiences.

The present study examined the relationship between immigrant status and successful aging and found that immigrants were significantly less likely to be aging successfully at time 2 compared to those born in Canada (65.6% vs. 71.6%; *x*^2^(1) = 20.6, *p* < 0.001), even though three-quarters of immigrant older adults (77.2%) had migrated to Canada for 40 years or more. It is consistent with findings that “healthy immigrant effect” disappears as the length of immigration increases [29]. Further research can explore if this is attributed to accumulated stress and inequalities experienced by immigrants over time. The multivariable binary logistic regression analysis results showed that Canadian-born older adults had approximately 24% higher odds of aging successfully after adjustment for 20 additional factors (aOR = 1.24, 95% CI: 1.09, 1.41).

To examine what baseline factors attenuate the association between immigrant status and successful aging, six different clusters of factors were added sequentially to the model. However, Canadian-born older adults always had higher odds of achieving successful aging (22.1% to 23.8%) compared to immigrant older adults, regardless of what factors were added. It appears that the factors included in the different models failed to attenuate the association between immigrant status and successful aging. Other factors in the full sample, in addition to immigrant status, which were significantly associated with successful aging included being younger, female sex, having higher income (i.e., above poverty line income), being married, not being obese, never smoking, engaging in moderate or strenuous physical activities, not having sleeping problems and living without heart disease or arthritis.

As there is no consistent definition of successful aging and how it can be measured [62,63], few studies on successful aging in Canada (with some exceptions, please see [4,5]) and no studies on successful aging among immigrants and refugees, it is difficult to develop direct parallels with the extant literature. Sadarangani [64] found that newly arrived immigrants in the United States who experienced acculturative stress, financial strain and limited English proficiency were more likely to develop poor health outcomes. The systems had ignored these elderly immigrants, so they were “aging out of place” [64] (p. 110). Although most immigrants in this study had migrated to Canada more than 40 years ago and therefore were not likely to be in the midst of adjustment problems, they might have other unmet needs for successful acculturation such as financial issues, language difficulties, ethnic discrimination and social isolation [65], contributing to a lower rate of successful aging than among their Canadian-born counterparts.

Consistent with the findings from previous research studies on potential factors associated with successful aging, the findings of this study reported that the odds of achieving successful aging were higher among older adults who were younger [4,66], as well as those who were married [4,67]; had higher income [68]; who were not obese [69,70]; who engaged in moderate to vigorous exercises [4,71,72,73]; who did not have sleeping problems [74]; and who reported better physical and mental health at baseline [4,5,75] This study showed that women had higher odds of achieving successful aging than men. In contrast, previous research findings had reported that men had higher odds of achieving successful aging than women [76].

Using the expanded definition of successful aging, the present study also found four possible outcomes of successful aging among all 18,978 respondents who met our inclusion criteria to be aged 60 years or older at time 2: (1) respondents who did not achieve successful aging at both baseline and time 2 (43.9%, 95% CI: 0.43, 0.45); (2) respondents who did not achieve successful aging at baseline but became successful agers at time 2 (11.7%, 95% CI: 0.11, 0.12); (3) respondents who achieved successful aging at baseline but became typical agers at time 2 (13.2%, 95% CI: 0.13, 0.14); and (4) respondents who achieved successful aging at both baseline and time 2 (31.1%, 95% CI: 0.30, 0.32). Therefore, successful agers may become typical agers over time, while typical agers may also become successful agers with time, possibly by engaging in activities that promote physical wellness, psychological and mental wellness, social wellness and self-rated wellness as described in the new measure of successful aging presented in this study. Additional research can examine these discrepancies to understand why some people live long and age well while some do not.

### 4.2. Limitations

The findings of this study should be interpreted in the context of the following limitations: Firstly, as a study using secondary data analysis to understand successful aging, the construction of the successful aging status variable is limited to the variables available in the CLSA data. For example, particular questions salient to the present study were not included in the CLSA, such as “do you have suicidal thoughts?” and questions related to Keyes’ concept of complete mental health (e.g., questions about aspects of psychological well-being such as “self-acceptance,” “personal growth,” “purpose in life,” “environmental mastery,” and “autonomy,” and questions about aspects of emotional well-being such as frequency of feeling “calm and peaceful” and “full of life”) [44] (p. 211) and questions related to spirituality (e.g., “a relationship with God or some higher power is important to me”) and gero-transcendence (e.g., “I think of my loved ones who have passed away and feel close to them”) [77] (p. 224) [77,78,79,80]. Secondly, according to Statistics Canada [81], the visible minority older adult population has increased from 2% in 1981 to 8.1% in 2011. In the CLSA, approximately 3.6% were visible minority members, slightly lower than the national average. Due to the small sample size, it is impossible to examine the relationship between successful aging and immigrants of diverse ethnic backgrounds. Thirdly, slightly less than half of Canadians aged 65 years and over (45%) had a post-secondary certificate, diploma, or degree, according to the 2016 Census [82]. However, respondents of the CLSA are very well-educated, with four in every five participants having a post-secondary degree or diploma (79.5%). Fourthly, the CLSA was conducted in English or French, excluding some of the most vulnerable immigrants who cannot communicate in either of Canada’s official languages. It is possible that racialized respondents, those with less education and those who cannot speak English or French, may face more systemic problems and have a lower incidence of successful aging. Fifthly, the present study is not able to examine the cohort effects of immigrant respondents and the impact of their migration history on successful aging as only two waves of the CLSA data were available at the beginning of this study. Future studies can fully exploit the longitudinal nature of the CLSA data and continue to examine the relationships between successful aging and various factors when more waves of the CLSA data have become available. Despite these limitations, the analyses of baseline and time 2 data of the CLSA provide valuable information on how people change over time and shed light on what baseline factors may attenuate the relationship between successful aging and immigrant status at time 2.

## 5. Conclusions

Successful aging is the core concept of the study of aging and the goal both of older adults and those who have devoted their lives to the care of older adults. The present study found that the prevalence of successful aging was significantly higher among Canadian-born older adults compared to immigrant older adults, even though three-quarters of the immigrants in this study had migrated to Canada 4 or more decades ago. Culturally and linguistically appropriate programs and services (e.g., acculturation programs, financial aids, language programs, information and referral services) may support immigrant older adults to age well. This study also found other significant baseline factors associated with successful aging included being younger, female sex, having higher income (i.e., above poverty line income), being married, not being obese, never smoking, engaging in moderate to strenuous physical activities, not having sleeping problems and living without heart disease, or arthritis. These findings are significant as many of these lifestyle, or health-related factors can be modified by encouraging older adults to engage in an active and healthy lifestyle (e.g., fitness classes, nutrition education); preventing chronic diseases (e.g., heart health programs, exercise programs for arthritis) and physical disabilities (e.g., fall prevention programs, road safety education); and encouraging positive mentality (e.g., programs to promote complete mental health and/or positive psychology). Policies and interventions focusing on these areas will support all older adults, Canadian-born and immigrant older adults, to achieve successful aging.

## Figures and Tables

**Figure 1 ijerph-19-13199-f001:**
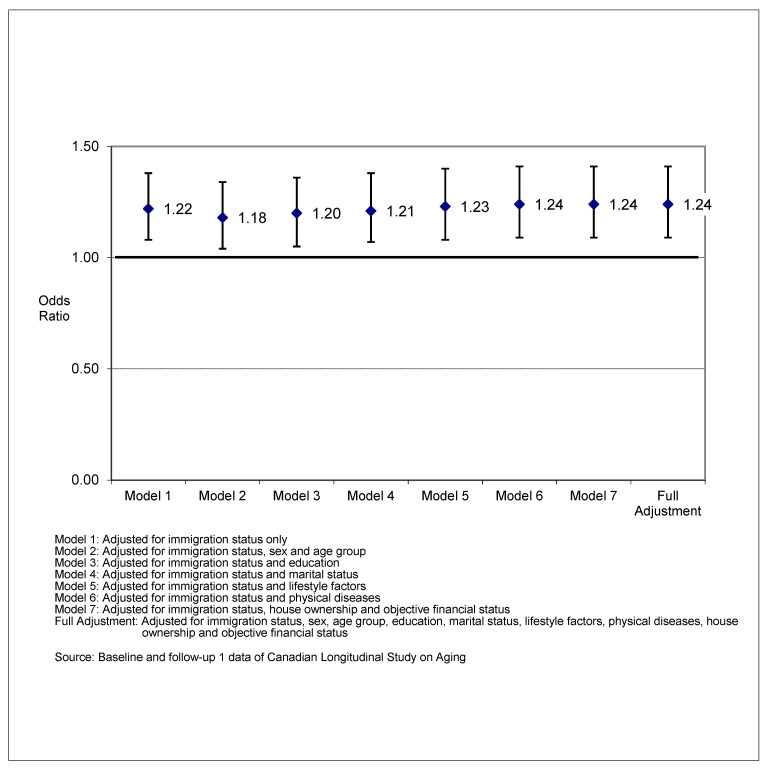
Adjusted odds ratio and 95% confidence interval of successful aging among Canadian-born older adults vs. immigrant older adults (*n* = 7651).

**Table 1 ijerph-19-13199-t001:** Description of elements of successful aging at both baseline and time 2.

Variable	Definition
Limitations in Activity of Daily Living (ADL)	Based on yes/no response to five questions on (1) ability to dress and undress oneself without help; (2) ability to eat without help; (3) ability to walk without help; (4) ability to walk with some help from a person or with the use of a walker or crutches, etc.; (5) ability to get in and out of bed without any help or aids. Coded as “no” if answered “no” to all of the five questions.
Limitations in Instrumental Activity of Daily Living (IADL)	Based on yes/no response to eight questions on (1) ability to use the telephone without help; (2) ability to get to places out of walking distance without help; (3) ability to go shopping for groceries or clothes without help; (4) ability to prepare own meals without help; (5) ability to do housework without help; (6) ability to do housework with some help; (7) ability to take own medicine without help; (8) ability to handle own money without help. Coded as “no” if answered “no” to all of the eight questions.
Disabling pain or discomfort	Derived from responses to two questions that asked if respondents were usually free of pain or discomfort (yes/no) and the number of activities prevented by the pain or discomfort (none, a few, some, most). Coded as “no” if “free from pain or discomfort” and “none or a few activities prevented by the pain or discomfort.”
Mental disorders—Anxiety, Depression, Posttraumatic Stress Disorder (PTSD)	Derived from responses to three questions that were (1) based on yes/no response to a question that asked if respondents had ever been told by a doctor that they had an anxiety disorder such as a phobia, obsessive-compulsive disorder or a panic disorder; (2) based on the Centre for Epidemiological Studies Short Depression Scale (CES-D 10) score, coded as not having depression if the CES-D 10 score < 10 [56,57]; (3) coded as not having PTSD based on the Primary Care Posttraumatic Stress Disorder (PC-PTSD) score < 3 [58].
Memory problems	Based on a yes/no question that asked if respondents had ever been told by a doctor that s/he had a memory problem. Coded as “no” if the respondents answered “no.”
Low mood—Felt depressed, felt happy, felt satisfied with life	Derived from responses to three questions that asked how often respondents felt depressed; felt happy; and felt satisfied with life [59] (all of the time (5–7 days), occasionally (3–5 days), some of the time (1–2 days), rarely or never (less than 1 day)). Coded as “no” if the respondents answered, “felt depressed rarely or never, or some of the time,” “felt happy occasionally or all of the time,” and “felt satisfied with life occasionally or all of the time.”
Lack of social support	Derived from responses to three questions that asked if respondents had (1) someone to give advice about a crisis; (2) someone who showed love and affection; (3) someone to confide in or talk to about oneself or one’s problems (none of the time, a little of the time, some of the time, most of the time, all of the time) [60]. Coded as “no” if the respondents answered “most of the time or all of the time” in all three questions.
Lack of self-rated wellness	Derived from responses to three questions that asked respondents to rate their (1) aging process; (2) perception of physical health; and (3) perception of mental health. Coded as “no” if the respondents answered “good to excellent” in all three questions.
Successful aging	Derived from responses to yes/no questions on (1) Limitations in ADL; (2) Limitations in IADL; (3) Disabling pain or discomfort; (4) Memory disorders; (5) Memory problems; (6) Low mood; (7) Lack of social support; (8) Lack of self-rated wellness. Coded as “yes” if the respondents answered “no” to all of these questions.

Independent variables assessed at the baseline wave of data collection.

**Table 2 ijerph-19-13199-t002:** Calculation of Poverty Line Status.

		Household Size										Total
		1 Person	2 Persons	3 Persons	4 Persons	5 Persons	6 Persons	7 Persons	8 Persons	9 Persons	10 Persons	
Household Income	<$20,000	756	184	44	16	7	8	0	0	0	2	1017
	$20,000–$49,999	2134	2123	322	108	38	20	1	2	0	0	4748
	$50,000–$99,999	1293	4525	676	238	76	30	8	4	2	0	6852
	$100,000–$149,999	266	2032	403	195	36	12	3	0	3	0	2950
	>$150,000	119	1329	356	171	45	10	6	0	0	0	2036
	Don’t know	254	241	40	14	5	3	4	1	0	0	562
	Refused	200	485	73	30	9	3	0	0	0	0	800
Total		5022	10,919	1914	772	216	86	22	7	5	2	18,965
**Legend**												
	Under poverty line income											
	Marginal income											
	Above poverty line income											
	Not answered											

**Table 3 ijerph-19-13199-t003:** Description of sample characteristics by successful aging status (*n* = 7651).

Variables	Successful Agers	Typical Agers	*x*^2^ (*df*), *p*-Value	% of Successful Agers
**Immigrant Status**				
Non-immigrant	4443 (82%)	1762 (78%)	20.6 (1), *p* < 0.001	72%
Immigrant	948 (18%)	498 (22%)		66%
**Sex**				
Male	2774 (52%)	1161 (51%)	0.005 (1), *p* = 0.946	71%
Female	2617 (49%)	1099 (49%)		70%
**Age groups** (years at baseline)				
55–59	879 (16%)	285 (13%)	189.7 (5), *p* < 0.001	76%
60–64	1630 (30%)	449 (22%)		77%
65–69	1232 (23%)	476 (21%)		72%
70–74	771 (14%)	351 (16%)		69%
75–79	613 (11%)	405 (18%)		60%
80+	266 (5%)	244 (11%)		52%
**Education**				
<Secondary school graduation	186 (4%)	127 (6%)	24.4 (2), *p* < 0.001	60%
Secondary school graduate and/or with some post-secondary education	836 16%)	390 (17%)		68%
Post-secondary degree/Diploma	4369 (81%)	1743 (77%)		72%
**Mortgage**				
Paying rent	585 (11%)	363 (16%)	40.5 (2), *p* < 0.001	62%
Paying mortgage	1164 (22%)	478 (21%)		71%
Paid off mortgage	3642 (68%)	1419 (63%)		72%
**Poverty line status**				
Under poverty line income	88 (2%)	85 (4%)	101.4 (3), *p* < 0.001	51%
Marginal income	932 (17%)	553 (25%)		63%
Above poverty line income	4047 (75%)	1461 (65%)		74%
No answer	324 (6%)	161 (7%)		67%
**Marital status (at baseline)**				
Single, never married or never lived with a partner	219 (4%)	137 (6%)	68.0 (3), *p* < 0.001	62%
Married	4273 (79%)	1606 (71%)		73%
Widowed	413 (8%)	274 (12%)		60%
Divorced/Separated	486 (9%)	243 (11%)		67%
**BMI**				
Underweight/Normal weight	1745 (32%)	645 (29%)	25.1 (2), *p* < 0.001	73%
Overweight	2403 (45%)	978 (43%)		71%
Obese	1243 (23%)	637 (28%)		66%
**Smoking status**				
Never smoked	1810 (34%)	708 (31%)	10.4 (2), *p* < 0.01	72%
Former smoker	3393 (63%)	1442 (64%)		70%
Current smoker	188 (4%)	110 (5%)		63%
**Sitting activity**				
Never/Seldom	75 (1%)	39 (2%)	1.21 (1), *p* = 0.271	66%
Sometimes/Often	5316 (99%)	2221 (98%)		71%
**Walking**				
Never/Seldom	1302 (23%)	639 (28%)	14.3 (1), *p* < 0.001	67%
Sometimes/Often	4089 (76%)	1621 (72%)		72%
**Light sports**				
Never/Seldom	4723 (88%)	2023 (90%)	5.54 (1), *p* < 0.02	70%
Sometimes/Often	668 (12%)	237 (11%)		74%
**Moderate sports**				
Never/Seldom	4979 (92%)	2144 (95%)	15.6 (1), *p* < 0.001	70%
Sometimes/Often	412 (8%)	116 (5%)		78%
**Strenuous sports**				
Never/Seldom	4092 (76%)	1859 (82%)	37.2 (1), *p* < 0.001	69%
Sometimes/Often	1299 (24%)	401 (18%)		76%
**Muscle & endurance exercises**				
Never/Seldom	4224 (78%)	1803 (80%)	1.94 (1), *p* = 0.164	70%
Sometimes/Often	1167 (22%)	457 (20%)		72%
**Sleep problem**				
Never/Rarely/Some of the time	4085 (76%)	1589 (70%)	24.8 (1), *p* < 0.001	72%
Occasional/All of the time	1306 (24%)	671 (30%)		66%
**Diabetes**				
No	4602 (85%)	1866 (83%)	9.54 (1), *p* < 0.005	71%
Yes	789 (15%)	394 (17%)		67%
**Heart disease**				
No	4848 (90%)	1928 (85%)	33.5 (1), *p* < 0.001	72%
Yes	543 (10%)	332 (15%)		62%
**Hypertension**				
No	3414 (63%)	1301 (58%)	22.4 (1), *p* < 0.001	72%
Yes	1977 (37%)	959 (42%)		67%
**Arthritis**				
No	4911 (91%)	2034 (90%)	2.29 (1), *p* = 0.131	71%
Yes	480 (9%)	226 (10%)		68%
**Osteoporosis**				
No	4873 (90%)	2009 (89%)	3.95 (1), *p* < 0.05	71%
Yes	518 (10%)	251 (11%)		67%

**Table 4 ijerph-19-13199-t004:** Adjusted odd ratios for successful aging based on binary logistic regression (*n* = 7651).

Variables	Immigrant Status Only	Immigrant Status + Age & Sex	Fully Adjusted
**Non-immigrant** (ref. immigrant)	**1.22 (1.08, 1.38)**	**1.18 (1.04, 1.34)**	**1.24 (1.09, 1.41)**
**Female** (ref. male)		1.04 (0.94, 1.15)	**1.17 (1.04, 1.31)**
**Age groups** (ref. 80+)			
55–59		**2.76 (2.19, 3.48)**	**2.16 (1.68, 2.79)**
60–64		**3.01 (2.42, 3.73)**	**2.50 (1.98, 3.16)**
65–69		**2.51 (2.01, 3.14)**	**2.20 (1.73, 2.79)**
70–74		**1.92 (1.52, 2.43)**	**1.73 (1.35, 2.21)**
75–79		1.45 (1.14, 1.85)	**1.36 (1.06, 1.74)**
**Education** (ref. < secondary school graduation)			
Secondary school graduate and/or with some post-secondary education			0.99 (0.76, 1.31)
Post-secondary degree/Diploma			1.05 (0.81, 1.36)
**Wealth Measure** (ref. paying rent)			
Paying mortgage			1.04 (0.86, 1.25)
Paid off mortgage			1.14 (0.97, 1.34)
**Poverty Line Status** (ref. under poverty line income)			
Marginal income			1.25 (0.90, 1.74)
Above poverty line income			**1.76 (1.27, 2.45)**
No answer			**1.58 (1.08, 2.30)**
**Marital status** (ref. single, never married or never lived with a partner)			
Married			**1.42 (1.11, 1.84)**
Widowed			1.28 (0.93, 1.75)
Divorced/Separated			1.26 (0.93, 1.69)
**BMI** (ref. obese)			
Underweight/Normal weight			**1.24 (1.07, 1.44)**
Overweight			**1.20 (1.05, 1.37)**
**Smoking status** (ref. current smoker)			
Never smoked			**1.46 (1.12, 1.89)**
Former smoker			1.37 (1.07, 1.77)
**Sitting activities** (ref. never/seldom)			1.37 (0.94, 2.00)
**Walking** (ref. never/seldom)			1.07 (0.95, 1.21)
**Light sports** (ref. never/seldom)			1.17 (0.99, 1.39)
**Moderate sports** (ref. never/seldom)			**1.35 (1.09, 1.67)**
**Strenuous sport** (ref. never/seldom)			**1.45 (1.27, 1.66)**
**Muscle or endurance exercises** (ref. never/seldom)			0.93 (0.81, 1.06)
**Sleep problem** (ref. occasionally/all of the time)			**1.29 (1.15, 1.44)**
**Diabetes** (ref. with condition)			1.14 (0.98, 1.31)
**Heart disease** (ref. with the condition)			**1.22 (1.04, 1.43)**
**Hypertension** (ref. with the condition)			1.03 (0.92, 1.15)
**Arthritis** (ref. with the condition)			**1.30 (1.08, 1.56)**
**Osteoporosis** (ref. with the condition)			1.15 (0.97, 1.37)

Remarks: Numbers in bold indicate that *p*-value < 0.05.

## Data Availability

Data are available from the Canadian Longitudinal Study on Aging (www.clsa-elcv.ca) for researchers who meet the criteria for access to de-identified CLSA data.

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
