# Peer review of "Successful Aging among Immigrant and Canadian-Born Older Adults: Findings from the Canadian Longitudinal Study on Aging (CLSA)"

_ijerph, 2022, doi:10.3390/ijerph192013199_

Round 1

Reviewer 1 Report

This is interesting work, as there are few studies on the crossover of diversity and successful ageing (SA). Countries with a high percentage of migrant population, such as Canada, need to advance in this type of study,

Overall the paper is well written, clear, methodologically sound and therefore coherent. I would only suggest addressing a few issues.

BACKGROUND

Successful Aging. It could include a broader, even holistic concept of SA. There are studies that include the dimensions of spirituality and gerotranscendence, I suggest you include in your work:

https://doi.org/10.3928/19404921-20110106-02

https://doi.org/10.1177/0020872819901147

https://doi.org/10.1093/geront/gnv127

https://doi.org/10.1093/gerona/glv099

Immigrant Status. Immigrant communities tend to cope well with ageing when they have strong social networks (family, neighbours) or are integrated into the community, especially through the religious community (church, community centre, etc.). ¿Podrán abordar esto en los antecedentes?

MATERIALS AND METHODS

The methodology is clear, well described and I only suggest the following:

Social Wellness. The study assesses social integration in a broader sense; social participation, religious worship attendance, integration in the neighbourhood.

DISCUSSION

The discussion, practical implications and limitations are correct.

Author Response

Response to Comments from the Reviewer #1

This is interesting work, as there are few studies on the crossover of diversity and successful ageing (SA). Countries with a high percentage of migrant population, such as Canada, need to advance in this type of study,

Overall the paper is well written, clear, methodologically sound and therefore coherent. I would only suggest addressing a few issues.

Thank you for your encouragement.

  1. BACKGROUND

Successful Aging. It could include a broader, even holistic concept of SA. There are studies that include the dimensions of spirituality and gerotranscendence, I suggest you include in your work:

https://doi.org/10.3928/19404921-20110106-02

https://doi.org/10.1177/0020872819901147

https://doi.org/10.1093/geront/gnv127

https://doi.org/10.1093/gerona/glv099

Thank you for your comment. Unfortunately, the CLSA does not provide information on spirituality and gerotranscendence, so we are unable to include this information in the analysis. This is highlighted in the limitations section:

“For example, particular questions salient to the present study were not included in the CLSA, such as "do you have suicidal thoughts?" and questions related to Keyes' concept of complete mental health (e.g., questions about psychological well-being such as "self-acceptance," "personal growth," "purpose in life," "environmental mastery," and "autonomy," and questions about emotional well-being such as frequency of feeling "calm and peaceful" and "full of life") [44] (p.211), and questions related to spirituality (e.g., “a relationship with God or some higher power is important to me”) and gerotranscendence (e.g., “I think of my loved ones who have passed away and feel close to them”) [77] (p.224) [77-80].” (p.15)

  1. Immigrant Status. Immigrant communities tend to cope well with ageing when they have strong social networks (family, neighbours) or are integrated into the community, especially through the religious community (church, community centre, etc.). ¿Podrán abordar esto en los antecedentes? [Will you be able to address this in the background?]

Thank you for your feedback. We have added this in the background section:

“Studies have found that older immigrants tend to do better if they have strong social networks (e.g., family, neighbors) [35,36], social participation (e.g., church or religious activities, community programs) [37,38], and access to the internet (e.g., social engagement, leisure activities) [39].” (p.3)

  1. MATERIALS AND METHODS

The methodology is clear, well described and I only suggest the following:

Social Wellness. The study assesses social integration in a broader sense; social participation, religious worship attendance, integration in the neighbourhood.

In keeping with a previous study of the CLSA on mental health as reference (i.e., Tong et al., 2020), social wellness was measured by the availability of social support

“Social wellness was measured by the availability of social support [60]. Respondents were asked if they had (1) someone to give them advice about a crisis; (2) someone to show them love and affection; and (3) someone to confide in or talk to about themselves or their problems. The response options were dichotomized into two levels: (1) none/a little/some of the time; and (2) most/all of the time. Respondents reporting most/all of the time in all three questions were considered as having social wellness.” (p.5)

In a future paper we will examine the relationship between specific types of social participation and successful aging, (e.g., church or religious activities, volunteer or charity work) as predictors for successful aging. This is outside the scope of the current paper, which already is quite long and includes many variables.

Tong, H., Lung, Y., Lin, S., Kobayashi, K.M., Davison, K.M., Agbeyaka, S., & Fuller-Thomson, E. (2020). Refugee status is associated with double the odds of psychological distress in mid-to-late life: Findings from the Canadian Longitudinal Study on Aging. Int. J. Soc. Psychiatry, 67(6), 747-760. https://doi.org/10.1177/0020764020971003

  1. DISCUSSION

The discussion, practical implications and limitations are correct.

Thank you for your time and comments.

Reviewer 2 Report

It has been a pleasure to review the manuscript “Successful aging among immigrant and Canadian-born older adults: Findings from the Canadian Longitudinal Study on Aging (CLSA)” (ijerph-1934564) submitted to the International Journal of Environmental Research and Public Health.

The paper explores the differences in successful ageing between nationals and migrants in Canada combining information of the first two waves of the Canadian Longitudinal Study on Aging. Using logit models, the authors find that the immigrants are less likely to achieve successful ageing than nationals and this result holds even after controlling for a wide set of covariates.

I think that the paper contributes to the literature of migrants’ health assimilation. That contribution is not so large, but it increases our knowledge of this process among older migrants. Furthermore, the evidence outside the U.S. is scant. Overall, the technical execution of the work is quite good. I find particularly valuable the following points:

— The discussion of why some chronic conditions might not automatically imply unsuccessful ageing.

— The use of different models that include very distinct sets of covariates. This makes clear that the results are robust and not driven by endogeneity problems.

— The systematic approach to successful ageing, extensively discussing the definition and providing solid arguments.

Regarding formal aspects, from my point of view, the organisation of the manuscript is appropriate and follows the scientific standards of the profession. It is technically sound, and tables and figures are self-explanatory and easy to understand.

Nevertheless, although I do not have major concerns about the quality of the paper, there are several comments and issues I would like to drop aiming to improve the paper:

(1) Probably, it is better not to use acronyms in the abstract but the complete form (e.g., ADL, IADL and CLSA).

(2) Although the paper is well written, I would try to reduce the use of the passive voice.

(3) When discussing the literature on the healthy immigrant effect, the authors should also refer to the relatively abundant literature on unhealthy assimilation of migrants, which is clearly in line with their results. In this respect, probably, the chapter of Antecol and Bedard (2015) in the Handbook of the Economics of International Migration is an excellent reference to consider.

(4) In subsection devoted to limitations, I think that the paper should mention that the present study, as it uses data in a cross-sectional way (even if it combines two waves of panel) is not able to take into account cohort effects (i.e., migrant arriving at different times might have different health status) and return migration (i.e., migrant with worse or better health could be more likely to come back to their country of origin). In the absence of panel data, the authors cannot address these issues, but, at least, they can mention them as limitations. Overall, these issues do not seem extremely relevant according to recent literature, but I would comment on them. The reference provided above (Antecol & Beard, 2015) is a survey that discuss this point. As a possible pathway for further work, I would suggest to consider the possibility of fully exploiting the longitudinal nature of the data when more waves are available to researchers.

(5) In the introduction, when motivating why the topic is relevant, I think it could be interesting to mention that there is evidence that suggest that households with disabled individuals need on average a higher level of income to achieve a certain level of well-being. The authors could cite several papers backing this point in the introduction (Antón et al., 2016; Banks et al., 2017).

(6) The inclusion of health and income variables is not exempt from problems, as limitations also affect the possibilities of achieving more income and improve the health status. The way the authors address this issue (using models with different covariates and show that the results are robust) is reasonably satisfactory. Nevertheless, I think that, when discussing the control covariates, the authors should explicitly comment on this (the potential endogeneity of some covariates and the subsequent use of different models with robust results).

References cited in the review

— Antecol, H., & Bedard, K. (2015). Immigrants and immigrant health. In B. R. Chiswick, & P. W. Miller (Eds.), Handbook of the Economics of International Migration (vol. 1A, 271–314). North Holland. https://doi.org/10.1016/B978-0-444-53764-5.00006-2

— Antón, J.-I., Braña, F. J., & Muñoz de Bustillo, R. (2016). An analysis of the cost of disability across Europe using the standard of living approach. SERIEs—Journal of the Spanish Economic Association, 7(3), 281–306 (2016). https://doi.org/10.1007/s13209-016-0146-5

— Banks, L. M., Kuper, H., & Polack, S. (2017). Poverty and disability in low- and middle-income countries: A systematic review. PloS ONE, 12(12). e0189996. https://doi.org/10.1371/journal.pone.0189996

Author Response

Response to Comments from the Reviewer #2

It has been a pleasure to review the manuscript “Successful aging among immigrant and Canadian-born older adults: Findings from the Canadian Longitudinal Study on Aging (CLSA)” (ijerph-1934564) submitted to the International Journal of Environmental Research and Public Health.

The paper explores the differences in successful ageing between nationals and migrants in Canada combining information of the first two waves of the Canadian Longitudinal Study on Aging. Using logit models, the authors find that the immigrants are less likely to achieve successful ageing than nationals and this result holds even after controlling for a wide set of covariates.

I think that the paper contributes to the literature of migrants’ health assimilation. That contribution is not so large, but it increases our knowledge of this process among older migrants. Furthermore, the evidence outside the U.S. is scant. Overall, the technical execution of the work is quite good. I find particularly valuable the following points:

— The discussion of why some chronic conditions might not automatically imply unsuccessful ageing.

— The use of different models that include very distinct sets of covariates. This makes clear that the results are robust and not driven by endogeneity problems.

— The systematic approach to successful ageing, extensively discussing the definition and providing solid arguments.

Regarding formal aspects, from my point of view, the organisation of the manuscript is appropriate and follows the scientific standards of the profession. It is technically sound, and tables and figures are self-explanatory and easy to understand.

Nevertheless, although I do not have major concerns about the quality of the paper, there are several comments and issues I would like to drop aiming to improve the paper:

  1. Probably, it is better not to use acronyms in the abstract but the complete form (e.g., ADL, IADL and CLSA).

Thank you for your excellent suggestion. We have changed all acronyms to their complete form in the abstract section:

“The concept of successful aging used in this study includes the ability to accomplish both activities of daily living (ADLs) and instrumental activities of daily living (IADLs), freedom from mental illness, memory problems and disabling chronic pain, adequate social support, and older adults' self-reported happiness, and subjective perception of their physical health, mental health, and aging process as good. Methods: The first two waves of data from the comprehensive cohort of the Canadian Longitudinal Study on Aging (CLSA) were analyzed.” (p. 1)

  1. Although the paper is well written, I would try to reduce the use of the passive voice.

Thank you for your feedback. We have edited the paper to reduce the use of the passive voice.

  1. When discussing the literature on the healthy immigrant effect, the authors should also refer to the relatively abundant literature on unhealthy assimilation of migrants, which is clearly in line with their results. In this respect, probably, the chapter of Antecol and Bedard (2015) in the Handbook of the Economics of International Migration is an excellent reference to consider.

Thank you for your comment. We have added the unhealthy assimilation of migrants in the background section:

“However, studies conducted in the United State have shown that this health advantage erodes over time and is replaced by the “unhealthy assimilation of immigrants” – the longer the immigrants stayed in the new countries, the poor their health status had becomes [32]. For example, the body mass index (BMI) also completely converged to native-born BMIs within 10 years of arrivals for women and 15 years of arrival for men [32].” (p.3)

  1. In subsection devoted to limitations, I think that the paper should mention that the present study, as it uses data in a cross-sectional way (even if it combines two waves of panel) is not able to take into account cohort effects (i.e., migrant arriving at different times might have different health status) and return migration (i.e., migrant with worse or better health could be more likely to come back to their country of origin). In the absence of panel data, the authors cannot address these issues, but, at least, they can mention them as limitations. Overall, these issues do not seem extremely relevant according to recent literature, but I would comment on them. The reference provided above (Antecol & Beard, 2015) is a survey that discuss this point. As a possible pathway for further work, I would suggest to consider the possibility of fully exploiting the longitudinal nature of the data when more waves are available to researchers.

Thank you for this excellent suggestion. It will be interesting to continue to study the relationships of successful aging and various factors when more waves of the CLSA data have become available. This is highlighted in the limitations section:

“Fifthly, the present study is not able to examine the cohort effects of immigrant respondents and the impact of their migration history on successful aging as only two waves of the CLSA data were available. Future studies can fully exploit the longitudinal nature of the CLSA data and continue to examine the relationships between successful aging and various factors when several more waves of the CLSA data have become available.” (p.15) 

  1. In the introduction, when motivating why the topic is relevant, I think it could be interesting to mention that there is evidence that suggest that households with disabled individuals need on average a higher level of income to achieve a certain level of well-being. The authors could cite several papers backing this point in the introduction (Antón et al., 2016; Banks et al., 2017).

Thank you for your comment. We have added this point in the introduction section:

“Studies also show that households of people with disabilities require, on average, a higher level of income to meet the same standard of living compared to households without people with disabilities [23], and there is a strong relationship between disability and poverty [24].” (p.2)

  1. The inclusion of health and income variables is not exempt from problems, as limitations also affect the possibilities of achieving more income and improve the health status. The way the authors address this issue (using models with different covariates and show that the results are robust) is reasonably satisfactory. Nevertheless, I think that, when discussing the control covariates, the authors should explicitly comment on this (the potential endogeneity of some covariates and the subsequent use of different models with robust results).

Thank you for your feedback. We have addressed this in the discussion section:

“To examine what baseline factors attenuate the association between immigrant status and successful aging, six different clusters of factors were added sequentially to the model. However, Canadian-born older adults always had higher odds of achieving successful aging (22.1% to 23.8%) compared to immigrant older adults, regardless of what factors were added. It appears that the factors included in the different models failed to attenuate the association between immigrant status and successful aging.” (p.14)

References cited in the review

— Antecol, H., & Bedard, K. (2015). Immigrants and immigrant health. In B. R. Chiswick, & P. W. Miller (Eds.), Handbook of the Economics of International Migration (vol. 1A, 271–314). North Holland. https://doi.org/10.1016/B978-0-444-53764-5.00006-2

— Antón, J.-I., Braña, F. J., & Muñoz de Bustillo, R. (2016). An analysis of the cost of disability across Europe using the standard of living approach. SERIEs—Journal of the Spanish Economic Association, 7(3), 281–306 (2016). https://doi.org/10.1007/s13209-016-0146-5

— Banks, L. M., Kuper, H., & Polack, S. (2017). Poverty and disability in low- and middle-income countries: A systematic review. PloS ONE, 12(12). e0189996. https://doi.org/10.1371/journal.pone.0189996

Thank you for your time and comments.